# Analysis of the Differential Relationship between the Perception of One’s Life and Coping Resources among Three Generations of Bedouin Women

**DOI:** 10.3390/ijerph16050804

**Published:** 2019-03-05

**Authors:** Orna Braun-Lewensohn, Sarah Abu-Kaf, Khaled Al-Said, Ephrat Huss

**Affiliations:** 1Conflict Management and Resolution Program, Department of Interdisciplinary Studies, Ben-Gurion University of the Negev, Beersheba 8410501, Israel; aks@bgu.ac.il (S.A.-K.); haled70@gmail.com (K.A.-S.); 2Kay Academic College of Education, Beersheba 84536, Israel; 3Social Work Department, Ben-Gurion University of the Negev, Beersheba 8410501, Israel; ehuss@bgu.ac.il

**Keywords:** Bedouin, women, coping resources, coping strategies

## Abstract

Bedouin society has undergone rapid changes over the past decade. The younger generation of Bedouin women is better educated, which has enabled them to enter different professions, increased their incomes and elevated their social status. We examined the sense of coherence (SOC) and its components of meaningfulness, manageability and comprehensibility as well as the use of coping strategies among Bedouin women from three age groups. We also investigated the coping resources and strategies before determining the relationships between these variables in the three groups. One hundred ninety-six women participated in the study. Differences were found mostly between the oldest age group (61 years and older) and the two younger groups (21–40 and 41–60 years old). The oldest women reported less meaningfulness and used less positive reframing, planning, humor and acceptance. In terms of coping strategies, venting was used more by the youngest group whereas behavioral disengagement was used more by the oldest group. In the younger groups, SOC and its components were positively correlated with the use of coping strategies that are considered to be adaptive and with emotional support. However, the correlations between these factors were negative among the oldest group, which points to non-adaptive coping strategies used by these women. These results are discussed in light of the salutogenic, stress-appraisal and coping theories.

## 1. Introduction

The Bedouin of the Negev are a minority group in Israel. They are Muslim Arabs who have inhabited the Negev desert since the fifth century Common Era (CE). Traditionally, they lived in nomadic or semi-nomadic tribes. However, over the past half century, they have experienced a rapid and dramatic transition. In the past decade alone, Bedouin society has undergone tremendous change, including the increased exposure to higher education and more interaction with Israeli Western society. These changes may have affected the coping resources of Bedouin women and their effectiveness in reducing stress [1]. Despite these changes, this society has preserved traditional customs and values, which emphasize the collective over the individual and collectivism remains a central component of Bedouin culture. In Bedouin culture, there is an emphasis on the mutual responsibility of the group for the individual that is accompanied by the cultural norms of respect for people in authority, privacy of feelings, politeness and belief in fate [2,3]. However, in recent years, Bedouin society has experienced significant changes [1]. The modernization and urbanization processes that the Bedouin community has undergone in recent years have disrupted traditional frameworks and led to the fragmentation of the cultural hierarchy. These changes have also highlighted the conflict between traditional Bedouin values and Western Israeli values, which are in direct competition with each other [4]. As a result, some of the society’s collectivist nature has been lost and this is reflected in the changes in traditional frameworks, the loss of traditional authority and the striving of young people for positions of leadership and individuality [5]. These changes have also influenced the women of this society. Bedouin women are now entering the higher-education system at dramatically higher rates. In 2018, the number of Bedouin women enrolled in the institutions of higher education was 50% higher than it was a decade earlier. However, this increased participation in higher education has not yet been completely translated into the workforce and most of these women still work in the education system [6].

The current study compared three generations of Bedouin women. Our goal was to compare these women in terms of the coping strategies that they use when faced with various stressful situations and their coping resources. Specifically, we focus of their sense of coherence (SOC) and its components of meaningfulness, manageability and comprehensibility.

### 1.1. Strategies for Coping with Stressful Events

The theoretical foundation of the present study is the transactional theory of stress and coping developed by Lazarus and Folkman [7]. They defined coping as the actual effort that is made in the attempt to make a perceived stressor more tolerable and minimize the distress induced by the situation. Most models of coping assume that the individuals who cope more effectively with stressful life events show lower levels of psychological problems and subsequently experience a better quality of life [7]. The importance of coping has been stressed in research conducted in recent decades. The studies of coping with stressful situations have shown that the emotionally focused strategies tend to be associated with more psychological problems whereas the use of problem-solving strategies or active coping tends to be linked to better wellbeing [8,9]. In addition to the increased stress that has accompanied the process of change in this society, the stress of aging should also be acknowledged. Getting older is stressful in itself. As individuals age, they experience social loss and general physical weakening, which can threaten their quality of life [10]. Since there are only limited studies focusing on the responses of Bedouin women to different types of stress, it is important to study coping strategies among Bedouin women as those strategies could have important implications for their wellbeing.

Several studies conducted in the last decade have pointed to the crosscultural importance of coping strategies. However, it seems that overall, among Arabs from various areas in Israel, similar to those living in other Western cultures, the implementation of problem-solving strategies acts as a protective factor whereas the implementation of emotional coping strategies seems to be maladaptive and results in more psychological problems and poorer wellbeing [11,12,13,14]. More specifically, among Bedouin women, seeking social support and regulating feelings and actions were found to be related to lower levels of wellbeing [15]. An exception for the well-known relationships between emotional coping and mental health was observed for the specific coping strategy of accepting responsibility. Although this strategy is considered to be a form of emotional coping, it was found to be negatively correlated with wellbeing among a group of Bedouin women [16].

### 1.2. Salutogenesis and Sense of Coherence (SOC)

Antonovsky [17] suggested the conceptualization of salutogenesis (the “origin of health”) in stress research and claimed that this is a universal concept that applies to all cultures. This is a continuum model that suggests that rather than classifying health/illness dichotomously, it is more appropriate to view each individual at any given moment as being somewhere on the ease/dis-ease continuum [18]. The most important concept in this model is SOC, which represents the motivation and the internal and external resources that one can use to cope with stressors. Thus, this SOC plays an important role in the way that one perceives challenges. Sense of coherence is a global orientation of seeing the world as more or less comprehensible (the internal and the external world are perceived as rational, understandable, consistent and expected), manageable (the individual believes that s/he has available resources to deal with situations) and meaningful (the motivation to cope and the commitment to emotionally invest in the coping process) [18].

The research conducted around the world, which has mostly been undertaken in Western societies, has found strong relationships between SOC and various health factors, including physical, mental and social functioning. Thus, an individual with a stronger SOC is expected to report better health [19]. In contrast to the strong conclusions regarding SOC and health in Western societies, some recent studies focusing on the responses to stressful events within Bedouin society have yielded different results. Although, SOC has been found to serve as a protective factor and to be negatively correlated with a range of psychological problems in some cases [20], SOC was not found to be related to mental health problems or was found to be positively related to such problems in most cases [21,22]. This complicated picture was presented in a recent study, which demonstrated that certain subgroups of Bedouin society benefit from SOC as a protective factor whereas SOC seems to be a maladaptive resource among other subgroups [23]. These results called for further investigation and the present research will try to fill part of this gap as it explores the relationships of SOC and its components with various coping strategies.

Several studies have investigated the relationships between SOC and coping strategies. One study of women with breast cancer in Sweden found that women who reported a strong SOC also reported using a variety of coping strategies. These women also reported using the coping strategies of distraction, situation redefinition, direct action, relaxation and to a lesser extent, religion. Using a variety of strategies enabled these women to choose the proper coping strategy and therefore, resulted in them coping better and reporting better psychological health [24]. A different study found that during a political violent event, both problem-solving and emotional coping strategies mediated the relationship between SOC and psychological problems. Although that study demonstrated a positive relationship between SOC and problem-solving, which was negatively correlated to psychological problems in turn, the relationships with emotional coping were reversed. SOC was negatively correlated with emotional coping, which was positively correlated with psychological problems in turn [25].

Following this literature review, we aim to explore several questions relating to SOC and coping strategies among Bedouin women. First, we would like to compare women from three age groups (young, middle-aged and older women) in terms of their SOC and its components of meaningfulness, manageability and comprehensibility, as well as their use of various coping strategies (i.e., self-distraction, active coping, emotional social support, instrumental social support, behavioral disengagement, venting emotions, positive reframing, planning, humor, acceptance and self-blame). Second, we would like to examine the relationships between SOC and the various coping strategies in the three age groups. Third, we aim to determine whether there are any differences in the direction or strength of these relationships among the different age groups.

## 2. Methods

### 2.1. Participants

One hundred ninety-six women who were aged 21–83 years old participated in this study. The mean age was 52.80 (*SD* = 16.82.) As for marital status, 22 women (11.9%) reported that they were single, 115 (62.2%) were a first wife, 40 (21.6%) were a second wife and 8 (4.3%) reported their family status as “other”. Furthermore, 11 women did not report their marital status. The years of formal education varied greatly among the women from 0–18 years, with a mean of 5.76 (*SD* = 5.77). As for current work, most of the women (62.9%) reported that they were not working at all at the time of the study.

The women were divided into three age groups. There were 49 women in the youngest group (21–40 years old), 73 women in the middle-aged group (41–60 years old) and 74 women in the oldest group (61–83 years old). The women in the youngest and middle-aged groups were more educated than the older women (*M* = 9.57 years, *SD* = 4.76; *M* = 8.18 years, *SD* = 5.56; *M* = 0.82 years, *SD* = 1.90, respectively; *F* = 76.14, *p* < 0.001). The middle-aged group included the most workers as 52.2% of the youngest group, 58.3% of the middle group and only 5.5% of the oldest group reported that they were currently working. However, both the youngest and the middle-aged groups differed significantly from the oldest group in terms of work (*X*^2^ = 49.90, *p* < 0.001).

### 2.2. Procedure

After receiving ethics approval from the university department’s IRB committee, the participants filled out anonymous self-report questionnaires between February and May 2018. The questionnaires were administered in Arabic, the native tongue of the participants. We used snowball, convenience sampling and adhered to all ethical guidelines. Participation was voluntary and the participants were informed that the researcher was interested in their experiences. Participants were free to withdraw their participation for any reason at any time during the questionnaire procedure.

### 2.3. Measures

#### 2.3.1. Demographic Background Data

The collected demographic data included age, marital status, years of formal education and working status.

#### 2.3.2. Brief COPE

This was a 28-item tool that measures coping strategies, using a four point Likert scale ranging from 1 (*usually don’t do it at all*) to 4 (*usually do it a lot*). The questionnaire is designed to fit different situations. The Brief COPE [26] items are divided into 14 subscales, which each contain two items. The means of each pair of items were used to create the subscales. Correlations were computed for each two items in order to determine reliability. Only the scales with reliability scores of *r* > 0.20 were included in the study. Thus, the following scales were used: self-distraction (*r* = 0.22), active coping (*r* = 0.22), emotional social support (*r* = 0.49), instrumental social support (*r* = 0.52), behavioral disengagement (*r* = 0.51), venting emotions (*r* = 0.26), positive reframing (*r* = 0.51), planning (*r* = 0.22), humor (*r* = 0.33), acceptance (*r* = 0.29) and self-blame (*r* = 0.31).

#### 2.3.3. Sense of Coherence

Sense of coherence [18] was measured using a series of semantic differential items that were rated on a 7-point Likert-type scale with anchoring phrases at each end. High scores indicated a strong SOC. An account of the development of the SOC scale and its psychometric properties appears in Antonovsky’s writings, which demonstrated that it was reliable and reasonably valid [17,18]. In this study, SOC was measured using the long-form scale consisting of 29 items. This scale includes items, such as: “*Doing the things you do every day is*” with answers ranging from 1 (*a source of pain and boredom*) to 7 (*a source of deep pleasure and satisfaction*). In the present study, we used the three subscales of SOC, which were namely meaningfulness, comprehensibility and manageability, and global SOC. The Cronbach’s alpha coefficient for the entire SOC scale was 0.82. The Cronbach’s alpha coefficients for the individual subscales were α = 0.81 for meaningfulness, α = 0.77 for comprehensibility and α = 0.79 for manageability.

### 3.3. Statistical Analyses

First, the frequencies, means and standard deviations of the demographic characteristics of the participants and the study variables were calculated. Second, in order to answer the first question, one-way ANOVA with LSD post hoc comparisons was conducted to understand the differences in the study variables among the three age groups. Third, we used Pearson correlations to explore the relationships among the study variables in each of the age groups. Finally, we computed Fisher *z*-scores to examine significant differences among these correlations in the three age groups.

## 3. Results

The preliminary results showed that for the entire sample, the SOC scores were quite low and there was little variance around the mean (*M* = 3.57). As for the coping strategies, it seems that this sample of women used self-distraction (*M* = 3.02) and self-blame (*M* = 3.13) as their main means of coping. The least commonly used coping strategies were emotional support (*M* = 2.06) and humor (*M* = 1.77).

To answer the first question, one-way ANOVA was conducted to compare the means of the different variables among the three age groups. The results of this analysis are presented in Table 1.

The results show that differences were most frequently seen between the youngest and the oldest groups, which were mainly in the coping-strategies variables. To a lesser extent, differences were observed between the middle-aged group and the oldest group. The differences between the youngest group and the middle-aged group were noted for only two scales of positive reframing and humor.

Pearson correlations were used to answer the second research question. For each age group, a correlation was run to explore the relationships between SOC and the coping strategies. The results of this analysis are presented in Table 2.

Our results revealed significant correlations between SOC and most of the coping strategies. As shown in Table 2, when we controlled for education, the direction, strength and significance of the correlations remained approximately the same in most cases. It should be noted that in the older age group, within which there was almost no variation in education, the correlations remained exactly the same. However, differences in the strength and the direction of the correlations were observed among the three age groups. Therefore, in order to understand whether there were significant differences between the groups in terms of the strength of these correlations, several *z*-tests were run to examine differences among the groups in the correlations between SOC and the different coping strategies. The results of this analysis are presented in Table 3.

In most cases (for all of the strategies, except for humor and acceptance), significant differences were observed. The most prominent differences were observed between the two younger groups, which were namely the youngest and the middle-aged group, and the oldest group. For emotional support, instrumental support, venting, positive reframing and planning, there were differences in the strength of the relationships and the direction of the relationships. In contrast, for active coping, behavioral disengagement and self-blame, the only differences were in the strength of the relationships. The only difference observed between the youngest group and the middle-aged group was in the strength of the relationship between the use of the coping strategy of planning and SOC, with a stronger relationship observed in the younger group.

To summarize, our results point to the fact that differences appeared mainly between the two younger groups and the oldest group. The youngest group and the middle-aged group tended to use the various coping strategies to the same extent and in most cases, they tended to make more use of coping strategies compared to the oldest women. Additionally, the relationships between SOC and the different coping strategies also differed mainly between the younger two groups and the oldest group. In contrast, among the oldest group, SOC was usually negatively correlated with the use of the different strategies. Among the youngest group and the middle-aged group, SOC was usually positively correlated with the use of adaptive strategies and negatively correlated with the use of emotional and non-adaptive strategies.

## 4. Discussion

This study examined Bedouin women from three age groups who represent different generations in a society that has undergone tremendous changes in recent years. Our aim was to explore similarities and differences in these women’s methods of coping and their coping resources. We also wanted to find out how the reported coping strategies and coping resources are related to each other in each of the examined groups. First, we examined some demographic background variables and as expected, we found that the older women were less educated than the younger women. Additionally, the older women were much less likely to work.

A preliminary examination revealed that all three groups reported very low SOC levels compared to those usually seen in Western societies and compared to the objective scale, with scores close to the mean of the scale. These results resemble those of other studies that have compared the Bedouin population with Jewish Israeli population (e.g., [27]). As Antonovsky [18] claimed, it could be that one has to belong to a society whose values and traditions are stable in order to develop a strong SOC. However, the transition that Bedouin society is going though [1] might result in and be reflected by the weak SOC exhibited by the participants in our study. The fact that our results show a trend of stronger SOC among the younger groups might indicate a minor shift in this society.

Regarding our first question, significant differences were found between the oldest women and the two younger groups. The meaningfulness component of SOC was strongest among the youngest women as it seems to be more accessible for them This indicates that for the younger generation, having meaning in one’s self and taking care of one’s self are seen to be more legitimate. On the other hand, the values on which the oldest women were raised and which were important for the former social structure of the Bedouin society are less prevalent among the younger generations. Thus, most of the values that the oldest women live for (e.g., strong relationships within the extended family, more power and respect for older people, very traditional and conservative sex roles) have been weakened and some of them no longer exist among the younger members of their community. The values that these women have lived for their entire lives are fading, leading to a decrease in the amount of respect granted to those who were formerly viewed as wise elders and a decrease in the amount of power held by those elders. Thus, this creates a situation in which these older women experience less meaningfulness in their lives. The same generational gap was observed in the behavioral components of the coping strategies. Indeed, planning, which has not traditionally been highly valued in the Bedouin context, was observed more among the members of the younger generations than among the members of the oldest generation. It seems that the younger generation is more aware of the importance of planning for the future, such as building a career and integrating that career with family life. In contrast, the oldest women believe that faith and religious values are the most important priorities. They believe that everything that happens is God’s will and therefore, there is no need to plan.

Differences in coping strategies were also observed for behavioral disengagement and venting. Behavioral disengagement was the only strategy that the oldest women used more than by their younger counterparts. It could be that the older women who are less educated and work less tend to have weak social status and therefore, they disengage more. To the extent that these older women believe that they have no control over outcomes and are at the mercy of luck or fate, they will most likely feel helpless and rely upon passive and avoidant coping strategies, such as behavioral disengagement. These women do not have a wide repertoire of coping skills to use so they lean on disengagement, which is more available to them. The other coping strategy for which we observed differences, venting, is used more by the youngest women. Traditionally, a ‘good’ woman does not complain or whine. It could be that women use this strategy more as they become more Westernized.

Our main aim in this study related to the relationships among the core concept of salutogenesis, sense of coherence (SOC)—a global orientation to see the world as more or less comprehensible, manageable and meaningful—and various coping strategies rooted in the stress, appraisal and coping theory; the problem-solving or emotional-coping orientation; and differences in these relationships among the different age groups. The oldest women seem to be embedded in the traditional collectivistic culture. Therefore, the role of SOC among these eldest women is different from the role of SOC among the youngest and the middle-aged groups. Our results indicate that among the oldest women, SOC is a protective factor as far as behaviors that are not adaptive among the oldest generation of the Bedouin population. Coping strategies, such as emotional and instrumental social support, do not help individuals from traditional collectivistic culture to overcome stressful events [16]. Thus, we can conclude that SOC first protects by preventing the use of non-adaptive behaviors. However, the role of SOC among the younger women who are becoming more Westernized appears to be more similar to the role of SOC in Western societies, in which SOC promotes good health [19]. Furthermore, SOC also seems to promote and advance adaptive behaviors among the younger women.

### Study Limitations

Several limitations should be acknowledged. First, all of the data were collected via self-report questionnaires. Therefore, all data are subjective. Second, a potential degree of sample bias cannot be ruled out as we investigated a relatively small sample (especially of the youngest age group), which was not a representative sample of Bedouin woman.

## 5. Conclusions

Based on the salutogenic theory and stress, appraisal and coping theories, we aimed to explore the role of coping resources and strategies when facing various stressful events. We explored this among three generations of Bedouin women who belongs to a society that has undergone rapid and dramatic changes during the past decade. We found some significant differences in the use of the various coping strategies and in the meaningfulness component of SOC among the three age groups. Moreover, we also found that SOC only prevents maladaptive behaviors for the oldest women while it also promotes the use of more adaptive coping strategies among the younger groups. It should be noted that the oldest women are an ‘at risk’ population as they do not make optimal use of coping strategies and SOC is a limited resource for them. These results give us, as researchers, an opportunity to demonstrate Antonovsky’s claim regarding the stability of SOC in a society that is undergoing tremendous changes. Further research should be conducted to expand our understanding of this matter. In the future, researchers might consider conducting a longitudinal study of the stability of SOC that would also consider variables, such as cultural change, aging, sociopolitical change and public-education achievements.

## Figures and Tables

**Table 1 ijerph-16-00804-t001:** Means and standard deviations of the study variables among the three age groups.

Variables	21–40 Years *N* = 49 (a) *M SD*	41–60 Years *N* = 68 (b) *M SD*	61–83 Years *N* = 74 (c) *M SD*	*F*
Sense of Coherence (SOC)	3.66 0.84	3.62 0.85	3.47 0.52	1.07
Comprehensibility	3.56 0.78	3.67 0.85	3.45 0.65	1.45
Manageability	3.54 0.93	3.57 0.88	3.50 0.54	0.12
Meaningfulness	3.93 1.18	3.61 0.98	3.46 0.69	3.34 *^(ac)^
Active coping	2.73 0.75	2.63 0.54	2.53 0.39	1.71
Emotional social support	2.05 0.96	2.09 0.82	2.04 0.59	0.05
Instrumental social support	2.58 0.59	2.52 0.55	2.46 0.58	0.61
Behavioral disengagement	2.52 0.88	2.71 0.89	2.95 0.67	4.30 *^(ac)^
Venting emotions	2.54 0.62	2.40 0.62	2.28 0.56	2.75 ᶺᶺ^(ac)^
Positive reframing	2.54 0.75	2.32 0.51	2.04 0.44	10.91 ***^(ab, ac, bc)^
Planning	2.61 0.63	2.57 0.43	2.36 0.49	4.13 *^(ac, bc)^
Humor	2.17 0.66	1.89 0.56	1.40 0.57	26.74 ***^(ab, ac, bc)^
Acceptance	2.74 0.75	2.59 0.63	2.48 0.61	2.43 ᶺ^(ac)^
Self-blame	3.06 0.74	3.11 0.64	3.19 0.67	0.54

ᶺ *p* = 0.09; ᶺᶺ *p* = 0.07; * *p* < 0.05; *** *p* < 0.001. *Note*: Venting (ac, *p* = 0.02); Acceptance (ac, *p* = 0.03).

**Table 2 ijerph-16-00804-t002:** Correlations between SOC and use of different coping strategies among the three age groups when we controlled for education.

	21–40 Years *N* = 49	21–40 Years, Controlling for Education	41–60 Years *N* = 68	41–60 Years, Controlling for Education	61–83 Years *N* = 74	61–83 Years, Controlling for Education
Active coping	0.69 ***	0.48 ***	0.61 ***	0.41 ***	0.00	0.00
Emotional support	0.65 ***	0.41 **	0.67 ***	0.48 ***	−0.33 **	−0.34 **
Instrumental support	0.07	−0.05	0.12	0.02	−0.51 ***	−0.51 ***
Behavioral disengagement	−0.70 ***	−0.54 ***	−0.64 ***	−0.51 ***	−0.41 ***	−0.41 ***
Venting	0.59 ***	0.34 *	0.36 ***	0.25 *	−0.34 **	−0.35 **
Positive reframing	0.67 ***	0.42 **	0.57 ***	0.42 ***	−0.13	−0.13
Planning	0.66 ***	0.40 **	0.26 *	0.18	−0.39 **	−0.39 **
Humor	0.13	0.00	0.27 *	0.11	0.13	0.14
Acceptance	−0.31 *	−0.13	−0.30 *	−0.13	−0.13	−0.14
Self-blame	−0.65 ***	−0.56 ***	−0.40 **	−0.33 **	−0.39 ***	−0.39 ***

* *p* ≤ 0.05; ** *p* ≤ 0.01; *** *p* ≤ 0.001.

**Table 3 ijerph-16-00804-t003:** Differences in the strengths of the correlations in the different age groups, as represented by *z*-scores.

	21–40 vs. 41–60	41–60 vs. 61–83	21–40 vs. 61–83
Active coping—SOC	0.72	30.68 ***	40.48 ***
Emotional support—SOC	−0.18	60.72 ***	50.91 ***
Instrumental support—SOC	−0.26	30.98 ***	30.34 ***
Behavioral disengagement—SOC	−0.57	−10.88 *	−20.28 *
Venting—SOC	10.56	40.26 ***	50.45 ***
Positive reframing—SOC	0.85	40.53 ***	40.97 ***
Planning—SOC	20.73 **	30.95 ***	60.36 ***
Humor—SOC	−0.76	0.85	0
Acceptance—SOC	−0.06	−10.04	−10.00
Self-blame—SOC	−10.83	−0.07	−10.92 *

* *p* ≤ 0.05; ** *p* < 0.01; *** *p* < 0.001.

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
