# Peer review of "Analysis of the Differential Relationship between the Perception of One’s Life and Coping Resources among Three Generations of Bedouin Women"

_ijerph, 2019, doi:10.3390/ijerph16050804_

Reviewer 1 Report

There are a couple of discussion points I would like to see expanded. First, lines 235-6, SOC in younger groups and a minor shift in society. Can you say more about this? This is an important point, especially in a rapidly changing society. Second, line 245 with "leads to less meaning in the lives of these older women". That seems quite an interpretive leap. How did you get there? Is there more that you can say?  I think more on those two points can add to this work and enhance its contribution to the literature.

Author Response

Thank you very much for giving us the opportunity to revise and resubmit our article. You will find our revised manuscript attached. We are grateful for all of the valuable comments. We have thoroughly reviewed all of them and considered each of them very seriously. We present our responses to each of the comments below.

Reviewer 1

There are a couple of discussion points I would like to see expanded.

First, lines 235-6, SOC in younger groups and a minor shift in society. Can you say more about this? This is an important point, especially in a rapidly changing society.

Answer: An explanation was added in response to this comment Lines 333-335.

Second, line 245 with "leads to less meaning in the lives of these older women". That seems quite an interpretive leap. How did you get there? Is there more that you can say?  I think more on those two points can add to this work and enhance its contribution to the literature.

Answer: We have tried to clarify and elaborate this point (Lines 340-343): “The values that these women have lived for their entire lives are fading, leading to a decrease in the amount of respect granted to those who were formerly viewed as wise elders and a decrease in the amount of power held by those elders, as well as to a situation in which these older women experience less meaningfulness in their lives.

To summarize, we made a number of changes to the article in response to the valuable comments we received. In addition, an English-speaking editor has edited the entire manuscript. We trust that these changes will satisfy the important concerns raised and hope that this revised paper is acceptable. We look forward to your reply.

Sincerely,                                                                                                                     

The authors

Reviewer 2 Report

We think text can improve and some aspects need some revision. From our point of view, there are three aspects authors should review:

1) The introduction does not give enough information and it is to much general. It is not clear if the text wants to talk about the specificity of Bedouin society or about Arab society in general. That question has to be clarified in the introduction and the introduction must be developed according to the objective defined. It is needed more precision about information it gives (for example, it talks about a “higher rate: 50%”, but it does not mention the year they are talking about neither where is that information from).

2) It is needed more explication about why is interesting to study the theory of stress and coping (Lazarus and Folkman) in relation to Bedouin women. The investigations mentioned by authors (about women with breast cancer in Sweden and another about israeli adolescents under missile attacks) talk about specific contexts that are very different from one they study. At the end, authors explain that have studied about the stability in a society that is undergoing tremendous changes. This idea should be developed at the beginning of the article.

3) The final discussion and conclusions repeat a lot the same idea (differences between oldest group and two younger group). It the study only analyses differences by age, the final conclusions are too poor. If authors really want to talk about the influence of Western in Arabs countries, it could be more useful to study differences by marital status, education level o work. Authors have this information from the sample but they decided to analyze only age. Why do they no try to understand  Bedouin women taking in account all the information they have of them?

Author Response

Thank you very much for giving us the opportunity to revise and resubmit our article. You will find our revised manuscript attached. We are grateful for all of the valuable comments. We have thoroughly reviewed all of them and considered each of them very seriously. We present our responses to each of the comments below.

Reviewer 2

We think text can improve and some aspects need some revision. From our point of view, there are three aspects authors should review:

1)     The introduction does not give enough information and it is to much general. It is not clear if the text wants to talk about the specificity of Bedouin society or about Arab society in general. That question has to be clarified in the introduction and the introduction must be developed according to the objective defined. It is needed more precision about information it gives (for example, it talks about a “higher rate: 50%”, but it does not mention the year they are talking about neither where is that information from).

Answer: The study focuses on Bedouin society. In the current version of the manuscript, we emphasize this point and have expanded the discussion of Bedouin society, in general, as well as the changes in Bedouin society during the last decade (lines 30-38).

“higher rate: 50%”- In this version, we include the year (2018). The source for this statistic [6] is the same as that of the information in the next sentence. Therefore, it is cited after the second sentence.

2)     It is needed more explication about why is interesting to study the theory of stress and coping (Lazarus and Folkman) in relation to Bedouin women. The investigations mentioned by authors (about women with breast cancer in Sweden and another about israeli adolescents under missile attacks) talk about specific contexts that are very different from one they study. At the end, authors explain that have studied about the stability in a society that is undergoing tremendous changes. This idea should be developed at the beginning of the article.

Answer: Studies on Bedouin women have been limited. Therefore, we referred to studies of populations that have some similarities to the examined one. Although there are some similarities between these populations, there are also differences. Thus, in the current version of the manuscript, we mention the fact that literature on the specific topic and population is limited and emphasize why this topic merits further study (lines 70-71).

3)     The final discussion and conclusions repeat a lot the same idea (differences between oldest group and two younger group). It the study only analyses differences by age, the final conclusions are too poor. If authors really want to talk about the influence of Western in Arabs countries, it could be more useful to study differences by marital status, education level o work. Authors have this information from the sample but they decided to analyze only age. Why do they no try to understand  Bedouin women taking in account all the information they have of them?

In accordance with the recommendations of Reviewers 2 and 3, we ran an analysis of the correlations in the three different groups in which we controlled for the level of education (since that was the only variable for which we had enough variation to conduct such an analysis). The results of that analysis confirmed the previous results (see Table 2 and Lines 207-210).

To summarize, we made a number of changes to the article in response to the valuable comments we received. In addition, an English-speaking editor has edited the entire manuscript. We trust that these changes will satisfy the important concerns raised and hope that this revised paper is acceptable. We look forward to your reply.

Sincerely,                                                                                                                     

The authors

Reviewer 3 Report

Due Feb 12, 2019

Review request: International Journal of Environmental Research and Public Health

Manuscript ID: ijerph-445387

Title:      Bedouin women: How do their coping resources relate to their coping strategies?

Synopsis

One hundred ninety-six Bedouin women in three age groups (ages 21 – 40, 41 – 60, 61 and above) were surveyed for their demographic information, sense of coherence (SOC), and coping strategies (Brief COPE) against stresses caused by rapid social and cultural change over the past decade. “The Bedouin community has undergone in recent years have disrupted traditional frameworks and lead to the fragmentation of the cultural hierarchy.” The meaningfulness of SOC along with stress coping strategies such as positive reframing, planning, humor and acceptance were less used by the older woman group than the younger groups. “Venting was used more by the youngest group; whereas behavioral disengagement was used more by the oldest group.” The correlation between SOC and the use of adaptive coping was positive for the youngest group, but negative for the oldest group. The discussion was then expanded with regard to salutogenic stress-appraisal and coping theories.

Reviewer's conflict of interest: None

Comments

This reviewer felt the discussion about the societal emphasis on collectivism versus individualism as a reflection of socialism/communalism versus capitalism insightful. “The conflict between traditional Bedouin values and Western Israeli values” compete with each other. Due to globalization, the changes in traditional and cultural values seems inevitable. These changes by themselves cannot be called the disruption or disintegration of traditionalism or conservativism, but simply a progression or evolution of society. This article does not make a one-sided argument. It is written objectively based on what the data show. The writing tone would most likely be changed if a qualitative or mixed-method research discussion was written from the perspective of ethnography or of critical theory, of which this study is not. Although the transactional model developed in the 1960s and 70s was reflected on a much slower pace of social change than the current rapid changes in our environment, the theoretical framework of salutogenesis and examination of coping strategies places this study in the right context. Overall, the article is a well-organized quantitative work with an appropriate theoretical foundation.

Detailed comments   

1.       Title. By definition, ‘resource’ is an action or strategy which may be adopted in adverse circumstances, but the title describes the relationship between ‘resources’ and ‘strategies’ which are the same concept. The SOC (meaningfulness, manageability, and comprehensibility) is not a direct measurement of resources but the perception about an individual’s life. It may be better to describe the study as the analysis of the differential relationship between the perception of life and coping resources (behaviors or strategies) among three generationally stratified Bedouin women. To be precise with terms, I recommend operationally defining each term.

2.       Abstract. It is more informative for readers if the size of the survey (n=196) is mentioned in the abstract. Operational definitions of terms used in the title can be explained here or in Introduction.

3.       Introduction: It should be noted that “aging” itself is a stressful event for women. The main aim of studying the different relationship to the perception of life and coping behaviors among three age-stratified samples was understandable; however, because educational levels were found to be confounded, additional analysis should control the educational level and measure the variables of interest (SOC and Brief COPE coping strategies) between the samples from different age groups. Do older women (age 61 and above) with high educational achievement display the same SOC and coping strategies as other women (with low educational level achievement) in the same age group? Is it possible that the older women with high educational achievement might have adopted to the cultural change similarly to younger women with high educational achievement?

4.       Methods: Please describe the methods for recruiting volunteers to participate. The methods of outreach should be fair and equitable and accordingly to ethical guidelines. The needs and rational for the study and its design are well presented. The 28-item Brief COPE appears to have different variations such as coping strategies on health or religious coping factors. It will be helpful for readers if you could add an appendix describing the 28 actual questions. 

5.       Results: Please be careful when using similar terms in the context of statistical analyses, for example the “significant association” (Line 192) tends to refer to the chi-square analysis while “the correlation” (Line 196) tends to refer to Pearson or Spearman correlation. In this section, I would appreciate seeing the results of additional analysis by controlling for education level. The analysis can also go in the other direction; by controlling the age, the study can address whether or not the difference in coping resources/life perception is due to education (stratified educational achievement). These analyses would be post-hoc, not a priori.

6.       Discussion: The reiteration of the main aim (Line 264) “(this study is) related to the relationships among SOC and the various coping strategies, as well as differences in these relationships among the different age groups” is clear; however, it would be clearer if the operational definition were given.

7.       Conclusion: The sentence (Line 285) “some substantial differences in the use of the various coping strategies, as well as in the meaningfulness component of SOC” is not clear. Please clarify how you quantified “substantial differences”. Line 289 “optimal use of coping strategies and SOC is a limited resource for them” is also not clear. How much would be limited, and how much would be a sufficient resource? Cultural changes, aging, sociopolitical movement, and public educational achievement are all confounding factors that can be used to investigate the stability of SOC. Please rephrase how you would propose to study the longitudinal stability of SOC in an individual. 

Overall, the project is well-conceived. As standardized instruments were used, providing operational definitions will help readers understand the research design, inquiry and conclusions. Additional statistical analyses will make the article more informative.  

End of review.

Author Response

Thank you very much for giving us the opportunity to revise and resubmit our article. You will find our revised manuscript attached. We are grateful for all of the valuable comments. We have thoroughly reviewed all of them and considered each of them very seriously. We present our responses to each of the comments below.

Reviewer 3

1.      Title. By definition, ‘resource’ is an action or strategy which may be adopted in adverse circumstances, but the title describes the relationship between ‘resources’ and ‘strategies’ which are the same concept. The SOC (meaningfulness, manageability, and comprehensibility) is not a direct measurement of resources but the perception about an individual’s life. It may be better to describe the study as the analysis of the differential relationship between the perception of life and coping resources (behaviors or strategies) among three generationally stratified Bedouin women. To be precise with terms, I recommend operationally defining each term.

Answer: The title was changed according to the reviewer’s suggestion.

2.      Abstract. It is more informative for readers if the size of the survey (n=196) is mentioned in the abstract. Operational definitions of terms used in the title can be explained here or in Introduction.

Answer: The suggested changes have been integrated into the abstract (Line 18).

3.      Introduction: It should be noted that “aging” itself is a stressful event for women. The main aim of studying the different relationship to the perception of life and coping behaviors among three age-stratified samples was understandable; however, because educational levels were found to be confounded, additional analysis should control the educational level and measure the variables of interest (SOC and Brief COPE coping strategies) between the samples from different age groups. Do older women (age 61 and above) with high educational achievement display the same SOC and coping strategies as other women (with low educational level achievement) in the same age group? Is it possible that the older women with high educational achievement might have adopted to the cultural change similarly to younger women with high educational achievement?

Answer: In accordance with the reviewer’s suggestion, in the present version of the manuscript, we note that aging is stressful for women (Lines 67-70). Also, as suggested, we ran the analysis again, this time controlling for level of education. As can be seen in Table 2, there was no variation in level of education among the older age group and, therefore, we couldn’t see any differences.

4.       Methods: Please describe the methods for recruiting volunteers to participate. The methods of outreach should be fair and equitable and accordingly to ethical guidelines. The needs and rational for the study and its design are well presented. The 28-item Brief COPE appears to have different variations such as coping strategies on health or religious coping factors. It will be helpful for readers if you could add an appendix describing the 28 actual questions.

Answer: We have described the methods used to recruit volunteer participants (Lines 146-147).

The version of the instrument that we used in the current version of the manuscript is cited in the text [26]: Carver, C.S. You want to measure coping but your protocol’s too long: Consider the brief COPE. Internatl. J. Behav. Med., 1997, 4(1), 92.

The division for the subscales is as follows: Self-Distraction, Items 1 and 19; Active Coping, Items 2 and 7; Denial, Items 3 and 8; Substance Use, Items 4 and 11; Use of Emotional Support, Items 5 and 15; Use of Instrumental Support, Items 10 and 23; Behavioral Disengagement, Items 6 and 16; Venting, Items 9 and 21; Positive Reframing, Items 12 and 17; Planning, Items 14 and 25; Humor, Items 18 and 28; Acceptance, Items 20 and 24; Religion, Items 22 and 27; and Self-Blame, Items 13 and 26. As mentioned in the text, we used only the reliable subscales. We do not think that it is necessary to include this information in the manuscript. However, if the reviewer would like us to add it, we will do so.

5.       Results: Please be careful when using similar terms in the context of statistical analyses, for example the “significant association” (Line 192) tends to refer to the chi-square analysis while “the correlation” (Line 196) tends to refer to Pearson or Spearman correlation. In this section, I would appreciate seeing the results of additional analysis by controlling for education level. The analysis can also go in the other direction; by controlling the age, the study can address whether or not the difference in coping resources/life perception is due to education (stratified educational achievement). These analyses would be post-hoc, not a priori.

Answer: The term association was removed and replaced with the term correlation, in accordance with the reviewer’s suggestion (Line 206).

In accordance with the reviewer’s recommendation, we ran an analysis of the correlations in the three different groups in which we controlled for education. The results of this analysis confirmed the previous results (see Table 2 and Lines 207-210).

6.       Discussion: The reiteration of the main aim (Line 264) “(this study is) related to the relationships among SOC and the various coping strategies, as well as differences in these relationships among the different age groups” is clear; however, it would be clearer if the operational definition were given.

Answer: Operational definitions were added (Lines 282-286).

7.       Conclusion: The sentence (Line 285) “some substantial differences in the use of the various coping strategies, as well as in the meaningfulness component of SOC” is not clear. Please clarify how you quantified “substantial differences”. Line 289 “optimal use of coping strategies and SOC is a limited resource for them” is also not clear. How much would be limited, and how much would be a sufficient resource? Cultural changes, aging, sociopolitical movement, and public educational achievement are all confounding factors that can be used to investigate the stability of SOC. Please rephrase how you would propose to study the longitudinal stability of SOC in an individual. 

Answer: Line 304 – “some substantial differences in the use of the various coping strategies, as well as in the meaningfulness component of SOC”… We have clarified this sentence by adding the phrase: “among the three age groups” (see also Table 1).

As suggested by the reviewer, we now note that future studies might address the longitudinal stability of SOC in individuals (Lines 313-316).

To summarize, we made a number of changes to the article in response to the valuable comments we received. In addition, an English-speaking editor has edited the entire manuscript. We trust that these changes will satisfy the important concerns raised and hope that this revised paper is acceptable. We look forward to your reply.

Sincerely,                                                                                                                      

The authors

Round  2

Reviewer 2 Report

Congratulations for your text!

I have only found some misprints: "that that" line 208 and "(NOT CLEAR)" line 257.

It should be revised.